

# Atmospheric River Tracking Method Intercomparison Project (ARTMIP): Project Goals and Experimental Design

Christine A. Shields[1], Jonathan J. Rutz[2], Lai-Yung Leung[3], F. Martin Ralph[4], Michael Wehner[5], Brian Kawzenuk[4], Juan M. Lora[6], Elizabeth McClenny[7], Tashiana Osborne[4], Ashley E. Payne[8], Paul Ullrich[7], Alexander Gershunov[4], Naomi Goldenson[9], Bin Guan[10], Yun Qian[3], Alexandre M. Ramos[11], Chandan Sarangi[3], Scott Sellars[4], Irina Gorodetskaya[12], Karthik Kashinath[13], Vitaliy Kurlin[14], Kelly Mahoney[15], Grzegorz Muszynski[13,14], Roger Pierce[16], Aneesh C. Subramanian[4], Ricardo Tome[11], Duane Waliser[17], Daniel Walton[18], Gary Wick[15], Anna Wilson[4], David Lavers[19], Prabhat[5], Allison Collow[20], Harinarayan Krishnan[5], Gudrun Magnusdottir[21], Phu Nguyen[22]

[1] Climate and Global Dynamics Division, National Center for Atmospheric Research, Boulder, CO, 80302, USA
[2] Science and Technology Infusion Division, National Weather Service Western Region Headquarters, National Oceanic and Atmospheric Administration, Salt Lake City, Utah, 84138, USA
[3] Earth Systems Analysis and Modeling, Pacific Northwest National Laboratory, Richland, Washington, 99354, USA
[4] Center for Western Weather and Water Extremes, Scripps Institution of Oceanography, La Jolla, California, 92037, USA
[5] Computational Chemistry, Materials, and Climate Group, Lawrence Berkeley National Laboratory, Berkeley, California, 94720, USA
[6] Department of Earth, Planetary, and Space Sciences, University of California, Los Angeles, California, 90095, USA
[7] Department of Land, Air and Water Resources, University of California, Davis, California, 95616, USA
[8] Department of Climate and Space Sciences and Engineering, University of Michigan, Ann Arbor, Michigan, 48109, USA
[9] Department of Atmospheric Sciences, University of Washington, Seattle, Washington, 98195, USA
[10] Joint Institute for Regional Earth System Science and Engineering, University of California, Los Angeles, California, 90095, USA
[11] Instituto Dom Luiz, Faculdade de Ciências, Universidade de Lisboa, 1749-016 Lisboa, Portugal
[12] Centre for Environmental and Marine Studies, University of Aveiro, 3810-193 Aveiro, Portugal
[13] Data & Analytics Services, National Energy Research Scientific Computing Center (NERSC), Lawrence Berkeley National Laboratory, Berkeley, California, 94720, USA
[14] Department Computer Science Liverpool, Liverpool, L69 3BX, UK
[15] Physical Sciences Division, Earth System Research Laboratory, National Oceanic and Atmospheric Administration, Boulder, CO, 80305, USA
[16] National Weather Service Forecast Office, National Oceanic and Atmospheric Administration, San Diego, CA, 92127, USA
[17] Earth Science and Technology Directorate, Jet Propulsion Laboratory, Pasadena, California, 91109, USA
[18] Institute of the Environment and Sustainability, University of California, Los Angeles, California, 90095, USA
[19] European Centre for Medium-Range Weather Forecasts, Reading, RG2 9AX, UK





[20]Universities Space Research Association, Columbia, MD, 21046, USA
[21]Department of Earth System Science, University of California Irvine, CA 92697, USA
[22]Department of Civil & Environmental Engineering, University of California Irvine, CA 92697, USA

*Correspondence to*: Christine A. Shields (shields@ucar.edu)

**Abstract**

The Atmospheric River Tracking Method Intercomparison Project (ARTMIP) is an
international collaborative effort to understand and quantify the uncertainties in atmospheric
river (AR) science based on detection algorithm alone. Currently, there are many AR
identification and tracking algorithms in the literature with a wide range of techniques and
conclusions. ARTMIP strives to provide the community with information on different
methodologies and provide guidance on the most appropriate algorithm for a given science
question or region of interest.  All ARTMIP participants will implement their detection
algorithms on a specified common dataset for a defined period of time. The project is divided
into two phases: Tier 1 will utilize the MERRA-2 reanalysis from January 1980 to June of
2017 and will be used as a baseline for all subsequent comparisons. Participation in Tier 1 is
required.  Tier 2 will be optional and include sensitivity studies designed around specific
science questions, such as reanalysis uncertainty and climate change. High resolution
reanalysis and/or model output will be used wherever possible. Proposed metrics include AR
frequency, duration, intensity, and precipitation attributable to ARs. Here we present the
ARTMIP experimental design, timeline, project requirements, and a brief description of the
variety of methodologies in the current literature. We also present results from our 1-month
"proof of concept" trial run designed to illustrate the utility and feasibility of the ARTMIP
project.



## 1 Introduction

Atmospheric rivers (ARs) are dynamically driven, filamentary structures that account for ~90% of poleward water vapor transport outside of the tropics, despite occupying only ~10% of the available longitude (Zhu and Newell 1998). ARs are often associated with extreme winter storms and heavy precipitation along the west coasts of mid-latitude continents, including the western US, western Europe, and Chile (e.g., Ralph et al., 2004;
Neiman et al., 2008; Viale and Nunez, 2011; Lavers and Villarini, 2015, Waliser and Guan, 2107). Their influence stretches as far as the polar caps as ARs transfer large amounts of heat and moisture poleward influencing the ice sheets surface mass and energy budget (Gorodetskaya et al., 2014; Neff et al., 2014; Bonne et al., 2015). Despite their short-term hazards (e.g., landslides, flooding), ARs provide long-term benefits to regions such as
California, where they contribute substantially to mountain snowpack (e.g. Guan et al. 2010), and ultimately refill reservoirs. The sequence of precipitating storms that often accompany ARs may also contribute to relieving droughts (Dettinger 2014). Because ARs play such an important role in the global hydrological cycle (Paltan et al 2017) as well as to water resources in areas such as the western US, understanding how they may vary from
subseasonal to interannual time scales and change in a warmer climate is critical to advancing understanding and prediction of regional precipitation (Gershunov et al., 2017).

The study of ARs has blossomed from 10 publications in its first 10 years in the 1990s to over 200 papers in 2015 alone (Ralph et al., 2017). This growth in scientific interest is
founded on the vital role ARs play in the water budget, precipitation distribution, extreme events, flooding, drought, and many other areas with significant societal relevance, and is evidenced by current (past) campaigns including the multi-agency supported CalWater

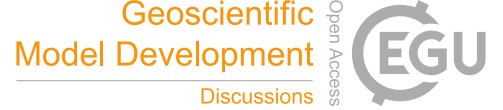

(Precipitation, Aerosols, and Pacific Atmospheric Rivers Experiment) and ACAPEX (ARM Cloud Aerosol Precipitation Experiment) field campaigns in 2015 with deployment of a wide range of in-situ and remote sensing instruments from four research aircraft, a research vessel, and multiple ground-based observational networks (Ralph et al., 2016; Neiman et al., 2017). The scientific community involved in AR research has expanded greatly, with 100+ participants from 5 continents attending the First International Atmospheric Rivers Conference in August 2016 (http://cw3e.ucsd.edu/ARconf2016/), many of whom enthusiastically expressed interest in the AR definition and detection comparison project described here.

The increased study of ARs has led to the development of many novel and objective AR identification methods for model and reanalysis data that build on the initial model-based method of Zhu and Newell (1998) and observationally-based methods of Ralph et al. (2004) and Ralph et al. (2013). These different methods have strengths and weaknesses, affecting the resultant AR climatologies and the attribution of high-impact weather and climate events to ARs. Their differences are of particular interest to researchers using reanalysis products to understand the initiation and evolution of ARs and their moisture sources (e.g., Dacre et al., 2015, Ramos et al., 2016a; Ryoo et al., 2015, Payne and Magnusdottir, 2016), to assess weather and subseasonal-to-seasonal (S2S) forecast skill of ARs and AR-induced precipitation (Jankov et al., 2009; Kim et al., 2013; Wick et al., 2013a; Lavers et al., 2014; Nayak et al., 2014 ; DeFlorio et al., 2017a, DeFlorio et al., 2017b; Baggett et al. 2017,), evaluate global weather and climate model simulation fidelity of ARs (Guan and Waliser, 2017), investigate how a warmer or different climate is expected to change AR frequency, duration, and intensity (e.g., Lavers et al., 2013; Gao et al., 2015; Payne and Magnusdottir, 2015; Warner et al., 2015; Shields and Kiehl, 2016 a/b; Ramos et al., 2016b; Lora et al.



2017; Warner and Mass, 2017), and attribute and quantify aspects of freshwater variability to ARs (Ralph et al., 2006; Guan et al., 2010; Neiman et al., 2011; Paltan et al., 2017).

Representing the climatological statistics of ARs is highly dependent on the identification method used (e.g., Huning et al., 2017). For example, different detection algorithms may produce different frequency statistics, not only between observation-based reanalysis products, but also among future climate model projections. The diversity of information on how ARs may change in the future will not be meaningful if we cannot understand how and why, mechanistically, the range of detection algorithms produce significantly different results. The variety of parameter variable types, and different choices that can be made for each variable in AR detection schemes is summarized in Fig. 1 and will be described in more detail in Section 3.

The detection algorithm diversity problem is not unique to ARs. For instance, the CLIVAR (Climate and Ocean – Variability, Predictability, and Change) program's IMILAST (Intercomparison of Mid Latitude Storm Diagnostics) project investigated extratropical cyclones similar to what is proposed here, (Neu et al., 2013). That project found considerable differences across definitions and methodologies and helped define future research directions regarding extratropical cyclones for such storms.  Hence, it is imperative to facilitate an objective comparison of AR identification methods, develop guidelines that match science questions with the most appropriate algorithms, and evaluate their performance relative to both observations and climate model data so that the community can direct their future work.

The American Meteorological Society (2017) glossary defines an atmospheric river as:

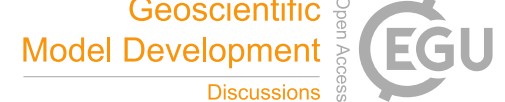

*"A long, narrow, and transient corridor of strong horizontal water vapor transport that is typically associated with a low-level jet stream ahead of the cold front of an extratropical cyclone. The water vapor in atmospheric rivers is supplied by tropical and/or extratropical moisture sources. Atmospheric rivers frequently lead to heavy precipitation where they are*

*forced upward—for example, by mountains or by ascent in the warm conveyor belt. Horizontal water vapor transport in the midlatitudes occurs primarily in atmospheric rivers and is focused in the lower troposphere".*

ARTMIP strives to evaluate each of the participating algorithms within the context of this

AR definition.

## 2   ARTMIP Goals

Numerous methods are used to detect ARs on gridded model or reanalysis data; therefore, AR characteristics, such as frequency, duration, and intensity, can vary substantially due to the chosen method. The differences between AR identification methods must be quantified and understood to more fully understand present and future AR processes, climatology, and impacts. With this in mind, ARTMIP has the following goals:

**Goal #1***: Provide a framework that allows for a systematic comparison of how different AR identification methods affect the climatological, hydrological, and extreme impacts attributed to ARs.*

The co-chairs and committee have established this framework by facilitating meetings, inviting participants, sharing resources for data and information management, and providing a common structure enabling researchers to participate. The experimental design, described in Section 4, is the product of the first ARTMIP workshop, and provides the framework necessary for ARTMIP to succeed. The final design was a collaborative

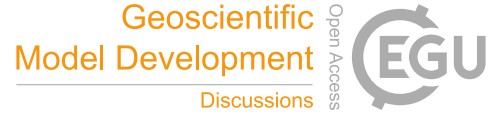

decision and included participation from researchers from around the world interested in a AR detection comparison project and who are co-authors on this paper.

**Goal #2:** *Understand and quantify the differences and uncertainties in the climatological*
*characteristics of ARs, as a result of different AR identification methods.*

The second goal is to quantify the extent to which different AR identification criteria (e.g., feature geometry, intensity, temporal, and regional requirements) contribute to the diversity, and resulting uncertainty, in AR statistics, and evaluate the implications to
understanding the thermodynamic and dynamical processes associated with ARs, as well as their societal impacts.

The climatological characteristics of ARs, such as AR frequency, duration, intensity, and seasonality (annual cycle), are all strongly dependent on the method used to identify ARs.
It is, however, the precipitation attributable to ARs that is perhaps most strongly affected, and this has significant implications for our understanding of how ARs contribute to regional hydroclimate now and in the future. For example, Fig. 2 highlights the results of three separate studies, (Dettinger et al., 2011, Rutz et al., 2014, Guan and Waliser, 2015), which used different AR identification methods to analyze the fraction of total cool-season
or annual precipitation attributable to ARs from a variety of reanalysis and precipitation datasets. Differences in AR identification methods as well as the techniques used to attribute precipitation to ARs have important implications for understanding the hydroclimate and managing water resources across the western US. For example, because so much of the western US water supply is accumulated and stored as snowpack during the
cool season, scientists and resource managers need to know how much of this water is attributable to ARs, and how changing AR behaviour might affect those numbers in the





future. The purpose of this figure is not to directly compare these analyses, but to motivate ARTMIP and illustrate the different ways of identifying and attributing precipitation that already exist in the literature. These results highlight the importance not only of quantifying the current uncertainty in AR climatology, but also the importance of future projections

and reliable estimates of their uncertainty.

**Goal #3**: *Better understand changes in future ARs and AR-related impacts.*

As a key pathway of moisture transport across the subtropical boundary and from ocean to

land, ARs are important elements of the global and regional water cycle. ARs also represent a key aspect of the weather–climate nexus as global warming may influence the synoptic-scale weather systems in which ARs are embedded and affect extreme precipitation in multiple ways. Hence, understanding the processes associated with AR formation, maintenance, and decay, and accurately representing these processes in climate models, is

critical for the scientific community to develop a more robust understanding of AR changes in the future climate. A key question that will be addressed is how different AR detection methods may lead to uncertainty in understanding the thermodynamic and dynamical mechanisms of AR changes in a warmer climate. Although the water vapor content in the atmosphere scales with warming following the Clausius-Clapeyron relation, changes in

atmospheric circulation such as the jet stream and Rossby wave activity may also have a significant impact on ARs in the future (Barnes et al., 2013, Lavers et al., 2015, Shields and Kiehl, 2016b). Will ARs from different ocean basins respond differently to greenhouse forcing? How do natural modes of climate variability come into play, i.e., El Nino–Southern Oscillation (ENSO), Arctic Oscillation (AO), Madden-Julien Oscillation (MJO),

the Pacific Decadal Oscillation (PDO), or the Southern Annular Model (SAM)? How do changes in precipitation efficiency influence regional precipitation associated with ARs in



the future? As the simulation fidelity of ARs is somewhat sensitive to model resolution (Hagos et al., 2015, Guan and Waliser, 2017), another important question is whether certain AR detection and tracking methods may be more sensitive to the resolutions of the simulations than others, and what are the implications to understanding uncertainty in projections of AR changes in the future.

To begin to answer and diagnose these questions, an understanding of how the definition and detection of an AR alters the answers to these questions is needed. A catalogue of ARs and AR-related information will enable researchers to assess which identification methods are most appropriate for the science question being asked, or region of interest. Applying different identification methods to climate simulations of ARs in the present day and future climate will facilitate more robust evaluation of model skill in simulating ARs and identification of mechanisms responsible for changes in ARs and associated extreme precipitation in a warmer climate. Finally, determination of the most appropriate methods of identifying ARs will provide for a set of best practices and community standards that researchers engaged in understanding ARs and climate change can work with and use to develop diagnostic and evaluation metrics for weather and climate models.

## 3 Method types

Table 1 summarizes the different algorithms adopted by the ARTMIP participants. Details for each parameter type and choice (from Fig. 1) are listed as table columns. The developer of the method is listed by row and refers to individuals or groups who developed the algorithm. The identifier in the first column (A1, A2, etc) will be used for Figs. 3, 5, 7, 8 and denotes those algorithms participating in the initial "proof of concept" phase of the project. Type choices are Condition or Track (see Section 3.1 for definition of these choices). Geometry





requirements refer to the shape and axis requirements, if any, of an AR object. For example, a "Condition" AR algorithm that tests a grid point may also have a requirement that strings grid points together to meet a minimum length, width, or axis. Threshold requirements refer to any absolute or relative threshold, typically for a moisture-related variable, that must be met

5    for an AR object to be defined. Temporal requirements refer to any time conditions to be met. Tracking algorithms typically contain temporal requirements to define an AR object as it is defined in time and space. However, many condition algorithms may also specify a minimum number of time instances (non-varying over a grid point) to be met before an AR object is counted for that grid point. Region refers to whether or not the algorithm is defined to track

10   or count ARs globally or only over specified regions. Reference lists published papers and datasets and their DOI numbers. "Experimental" algorithms have not been published yet.

**Table 1**. Algorithm methods participating in ARTMIP participants. Developer is listed along with algorithm details, i.e., type; geometry, threshold, and temporal requirements; region of study; DOI reference. Identifiers for the subset of methods participating in the one month

25   "proof of concept" test are in the far-left column and labeled as A1, A2, etc.





| | Developer | Type | Geometry Requirements | Threshold Requirements | Temporal Requirements | Region | DOI/ Reference |
|---|---|---|---|---|---|---|---|
| A1 | Gershunov et al.[+] | Condition and Track | >= 1500km long | Absolute: 250kgm$^{-1}$s$^{-1}$ IVT 1.5cm IWV | Time stitching -18 hours (3 time steps for 6 hourly data) | Western U.S. | 10.1002/ 2017GL0 74175 |
| A2 | Goldenson[+] | Condition | >2000km long and < 1000km wide, Object recognition | Absolute: 2cm IWV | Time slice | Western U.S. | Goldenso n et. al, In review |
| | Gorodetskaya et al. | Condition | IWV > thresh. at the coast (within defined longitudinal sector) and continuously at all latitudes for ≥20° equatorward (length > 2000 km), within ±15° longitude sector (width of 30° ~1000 km at 70°S; requirement of meridional extent) | Relative: *ZN using IWV adjusted for reduced tropospheric moisture holding capacity at low temperatures (AR$_{coeff}$= 0.2) | Time slice | Polar (East Antarctica) | 10.1002/ 2014GL0 60881 |
| A3 | Guan and Waliser[+^] | Condition | Length >2000km and length-width ratio>2; Coherent IVT | Relative: 85th percentile IVT; Absolute min requirement designed for | Time slice | Global | 10.1002/ 2015JD0 24257; Guan et al., 2017, |



| | | | | | | | |
|---|---|---|---|---|---|---|---|
| | | | direction within $45^0$ of AR shape orientation and with a poleward component | polar locations: $100 kgm^{-1}s^{-1}$ IVT | | | JHM, submitted |
| A4 | Hagos et al. [+] | Condition | Dependent on threshold requirements to determine footprint; > 2000 km long and < 1000 km wide | Absolute: 2cm IWV $10 ms^{-1}$ wind speed | Time slice | Western U.S. | 10.1002/ 2015GL0 65435;  10.1175/ JCLI-D- 16- 0088.1 |
| | Lavers et al. | Condition | $4.5^o$ latitude movement allowed | Relative: ~$85^{th}$ percentile determined by evaluation of reanalysis products | Time slice | UK, Western US | 10.1029/ 2012JD0 18027 |
| A5 | Leung and Qian[+] | Track | Moisture flux has an eastward or northward component at landfall; tracks originating north of 25N and east of 140W are rejected | Absolute: mean IVT along track > 500 $kgm^{-1}s^{-1}$ and IVT at landfall > 200 $kgm^{-1}s^{-1}$; grid points up to 500km to the north and south along the AR tracks are included as part of the AR if their mean IVT > 300 $kgm^{-1}s^{-1}$ | Time slice | Western U.S. | 10.1029/ 2008GL0 36445 |





| A6, A7 | Lora et al.[+] | Condition | Length >= 2000km | Relative: IVT 100kgm$^{-1}$s$^{-1}$ above climatological area means for N. Pacific | Time slice | Global (A6), North Pacific (A7) | 10.1002/ 2016GL0 71541 |
|---|---|---|---|---|---|---|---|
| | Mahoney et al. | Condition and Track | Length >= 1500km, Width <=1500km | Absolute: ARDT-IVT 500kgm$^{-1}$s$^{-1}$ for SEUS. | See Wick | Southeast U.S. | 10.1175/ MWR-D- 15- 0279.1 (uses Wick) |
| | Muszynski et al. | Condition | Topological analysis and machine learned | Threshold-free | N/A | Western U.S., adaptable to other regions | Experime ntal |
| A8 | Payne and Magnusdottir [+^] | Condition | Length > 1200km, landfalling only | Relative: 85$^{th}$ Percentile of maximum IVT (1000-500mb) Absolute: IWV >2cm, 850mb wind speed > 10m/s | Time stitching (12-hour minimum) - | Western U.S. | 10.1002/ 2015JD0 23586; 10.1002/ 2016JD0 25549 |
| | Ralph et al. | Condition | Length >= 2000km, Width <= 1000km | Absolute: IWV 2cm | Time slice | Western U.S. | doi:10.1 175/152 0- 0493(20 04)132< 1721: sacaoo> 2.0.co;2 |
| A9 | Ramos et al.[+^] | Condition | Detected for reference meridians, length >=1500km, latitudinal | Relative: IVT 85$^{th}$ percentile (1000-300mb) | Time slice, but 18-hour minimum for persistent ARs | Western Europe, South Africa, adaptable to other regions | 10.5194/ esd-7- 371-2016 |





| | | | | | | | |
|---|---|---|---|---|---|---|---|
| | | | movement <4.5$^0$N | | | | |
| A10 | Rutz et al.[+] | Condition | Length >= 2000km | Absolute: IVT (surface to 100mb) = 250kgm$^{-1}$s$^{-1}$ | Time slice | Global, low value on tropics | 10.1175/ MWR-D-13-00168.1 |
| A11, A12, A13 | Sellars et al.[+] | Track | Object identification | Absolute: IVT, thresholds tested = 300 (A11), 500 (A12), 700 (A13) kgm$^{-1}$s$^{-1}$ | Time stitching, minimum 24-hour period | Global | 10.1002/ 2013EO3 20001; 10.1175/ JHM-D-14-0101.1 |
| A14 | Shields and Kiehl[+] | Condition | Ratio 2:1, length to width grid points min 200km length; 850mb wind direction from specified regional quadrants, landfalling only | Relative: *ZN moisture threshold using IWV; Wind threshold defined by regional 85$^{th}$ percentile 850mb wind magnitudes | Time slice | Western U.S. Iberian Peninsula, UK, adaptable but regional specific | 10.1002/ 2016GL0 69476; 10.1002/ 2016GL0 70470 |
| A15 | TEMPEST[+] | Track | Laplacian IVT thresholds most effective for widths >1000km; cluster size minimum = 120000km$^2$ | IVT >=250kgm$^{-1}$s$^{-1;}$ | Time stitching | Global, but latitude >=15$^o$ | Experime ntal |



| Walton et al. | Condition and Track | Length >= 2000 km | Relative: IVT > 250 kg/m/s + daily IVT climatology | Time stitching, minimum 24-hour period | Western U.S. | Experimental |
|---|---|---|---|---|---|---|
| Wick et al. | Condition and Track | >=2000km long, <= 1000km wide object identification involving shape and axis | Absolute: ARDT-IWV >2cm | Time slice and stitching | Regional | 10.1109/ TGRS.20 12.22110 24 |

*ZN relative threshold formula: $Q >= Q_{zonal\_mean} + AR_{coeff} * (Q_{zonalmax} - Q_{zonamean})$ where Q = moisture variable, either IVT (kg m$^{-1}$s$^{-1}$) or IWV(cm). $AR_{coeff} = 0.3$ except where noted. (Zhu and Newell, 1998). The Gorodetskaya method uses $Q_{sat, where}$ $Q_{sat}$ represents maximum

moisture holding capacity calculated based on temperature (Clausius-Clapeyron), an important distinction for polar ARs.

+Methods used in a 1-month proof-of-concept test (Section 5). These methods are assigned an algorithm id, i.e. A1, A2, etc.

^ These 1-month proof-of-concept methods apply a percentile approach to determining ARs.

A3 and A8 applied the full MERRA2 climatology to compute percentiles. A9, applied the Feb 2017 climatology for this test only. For the full catalogues, A9 will apply extended winter and extended summer climatologies to compute percentiles.

**3.1 Condition vs tracking algorithms**

The subtleties in language when describing different algorithmic approaches are best illustrated with the "tracking" versus "condition" parameter type. For ARTMIP purposes, two



basic detection "types," defined at the first ARTMIP workshop, represent two fundamentally different ways of detecting ARs. "Condition" refers to counting algorithms that identify a time instance where AR conditions are met. Condition algorithms typically search grid point by grid point for each unique time instance. If AR geometry (involving multiple grid points)

and threshold requirements are met, then an AR "condition" is found at that grid point and that point in time. Condition methods may also have an added temporal requirement, but this does not impact the fact that conditions are met at a unique point in space (grid point).

"Tracking" refers to a Lagrangian-style detection method where ARs are objects that can be

tracked (followed) in time and space. Objects have specified geometric constraints and can span across grid points. Tracking algorithms must include a temporal requirement that stitches time instances together, i.e., a tracked AR would include several 3-hour time slices stitched together. An example of an object-oriented tracking methods is the Sellars et al., 2015 tracking method.

### 3.2 Thresholding: absolute versus relative approaches

Another major area where algorithms diverge is in how to determine the intensity of an AR. Some methods follow studies, such as Ralph et al. (2004) and Rutz et al. (2014), that assign an observationally-derived value, such as 2 cm of IWV, or an IVT value of 250 kg m$^{-1}$s$^{-1}$ to determine the physical threshold required for identification of an AR. Other methods use a statistical approach rather than an absolute value when setting a threshold value, such as the

approach developed by Lavers et al. (2012) where an AR is defined by the 85$^{th}$ percentile values of IVT (kg m$^{-1}$s$^{-1}$). Other relative threshold methods, such as Shields and Kiehl





(2016a/b), and Gorodetskaya et al. (2014), apply a direct interpretation of the foundational Zhu and Newell (1998) paper that defines ARs by computing anomalies of IWV (cm) or IVT $(kg\ m^{-1}s^{-1})$, by latitude band. Further, Gorodetskaya et al. (2014) used the physical approach to define a threshold for IWV depending on the tropospheric moisture holding capacity as a

function of temperature at each pressure level (Clausius-Clapeyron relation). The Lora et al. (2017) method is yet another relative thresholding technique wherein ARs are detected for IVT at 100 kg $m^{-1}s^{-1}$ above a climatological-derived mean IVT value and thus changes with the climate state. Although all of these methods "detect" ARs, they do not always detect the same object. Observationally based methods may be best for case studies, forecasts, or

current climatologies, but future climate research may be better served by relative methodologies, partly because of model biases in the moisture and/or wind fields. Ultimately, however, the best algorithmic choice will be unique to the science being done, rather than depending on general categories.

**4 Experimental Design**

ARTMIP will be conducted using a phased experimental approach. All participants must contribute to the first phase to provide a baseline for all subsequent experiments in the second phase. The first phase will be called Tier 1 and will require that participants provide a

catalogue of AR occurrences for a set period of time using a common reanalysis product. This phase will focus on defining the uncertainties amongst the various detection method algorithms. The second phase, Tier 2, is optional, and will potentially include creating catalogues for a number of common datasets with different science goals in mind. To some degree, the experiments chosen for Tier 2 will be informed by the outcomes of Tier 1;

however, initially, ARTMIP participants have proposed three separate Tier 2 experiments. The first experiment will explore the uncertainties to the various reanalysis products, and



second and third set of experiments will be testing AR algorithms under climate change
scenarios and different model resolutions.  Table 2 outlines the timeline for ARTMIP.

**Table 2**. ARTMIP Timeline. Completed targets are in bold.

| Target Date | Activity |
| --- | --- |
| **May 2017** | **1st ARTMIP Workshop** |
| **August/September 2017** | **1-Month Proof of Concept Test** |
| January 2018 | Full Tier 1 Catalogues Due |
| Winter 2017/2018 | Tier 1 Analysis and Scientific Papers |
| Spring 2018 | Tier 2 Catalogues Due |
| Spring 2018 | 2nd ARTMIP Workshop |
| Summer/Fall 2018 | Tier 2 Analysis and Scientific Papers |

## 4.1 Tier 1 description

ARTMIP participants will run their independent algorithms on a common reanalysis dataset
10  and adhere to a common data format. Tier 1will establish baseline detection statistics for all
participants by applying the algorithms to MERRA-2 (Modern Era Retrospective-Analysis
for Research and Applications, (Gelaro et al., 2017, Data DOI number:
10.5067/QBZ6MG944HW0) reanalysis data, for the period of January 1980 – June 2017. To
eliminate any processing differences between algorithm groups, all moisture and wind
15  variables have been processed and made available at the University of California, San Diego
(UCSD) Center for Western Weather and Water Extremes (CW3E) (B. Kawzenuk, personal
communication) at ~50km (.5$^{o}$ x .625$^{o}$) spatial resolution and 3-hourly instantaneous





temporal resolution. Specifically, ARTMIP participants that require IVT (integrated vapor transport, kg m$^{-1}$s$^{-1}$) information for their algorithms will be using IVT data calculated by UCSD using the MERRA-2 data 3-hourly zonal and meridional winds, and specific humidity variables. IVT is calculated using the following Eq. (1), (from Cordeira et al., 2013),

$$ \text{IVT} = -\frac{1}{g} \int_{Pb}^{Pt} (q(p) \, \boldsymbol{V}_h(p)) \, dp \qquad (1) $$

where q is the specific humidity (kg/kg), $\boldsymbol{V}_h$ is the horizontal wind vector (ms$^{-1}$), Pb is 1000 hPa, Pt is 200 hPa, and g is the acceleration due to gravity. The 1-hourly averaged IVT data

10 available from MERRA-2 directly will not be used. A comparison between 3-hourly UCSD IVT-computed data and 1-hourly MERRA-2 data was completed with details found in supplemental information. Although the 1-hour data provides better temporal resolution, the 3-hourly provides ample temporal information and is sufficient for algorithmic detection comparisons for ARTMIP. Gains using the 1-hourly MERRA-2 IVT data do not outweigh the

15 extra burden in computational resources required for groups to participate in ARTMIP.

Not all algorithms require IVT. Instead, some use IWV, integrated water vapor, or precipitable water (cm). This quantity is derived from MERRA-2 data and is computed as Eq. (2)

$$ \text{IWV} = -\frac{1}{g} \int_{Pb}^{Pt} q(p) \, dp \qquad (2) $$

where q is the specific humidity (kg/kg) , Pb is 1000 hPa, Pt is 200 hPa, and g is the acceleration due to gravity. Table 3 summarizes all the MERRA-2 data available for AR tracking.



**Table 3**. ARTMIP variables available for detection algorithms.

| Variable | Variable Units | Description | Level |
|---|---|---|---|
| U | $\text{ms}^{-1}$ | Zonal wind | All pressure levels |
| V | $\text{ms}^{-1}$ | Meridional wind | All pressure levels |
| Q | kg/kg | Specific humidity | All pressure levels |
| T | Kelvin | Air Temperature | All pressure levels |
| IVT | $\text{kg m}^{-1}\text{s}^{-1}$ | Integrated vapor transport | Integrated from 1000 to 200 hPa |
| IWV | mm | Integrated water vapor | Integrated from 1000 to 200 hPa |
| uIVT | $\text{kg m}^{-1}\text{s}^{-1}$ | Zonal wind component of IVT | Available as integrated or pressure level |
| vIVT | $\text{kg m}^{-1}\text{s}^{-1}$ | Meridional wind component of IVT | Available as integrated or pressure level |

Once catalogues are created for each algorithm, data will be made available to all
participants. Data format specifications for each catalogue are found in supplementary
material.



Many of the ARTMIP participants focus on the North Pacific (Western North America) and
North Atlantic (European) regions, however, ARs in other regions, such as the poles and the
Southeast U.S. may also be evaluated with ARTMIP data. We are not placing any coverage
requirements for participation in ARTMIP, and each group can provide as many global or
regional catalogues as desired.

**4.2 Tier 2 description**

Tier 2 will be similar in structure to Tier 1 in that all participants will create catalogues on a
common dataset and follow the same formats, etc.  However, instead of algorithms creating
catalogues for one reanalysis product, a number of sensitivities studies will be conducted
spanning AR detection sensitivity to reanalysis products, and AR detection sensitivity under
climate change scenarios.

**4.2.1 Reanalysis catalogues**

For the reanalysis sensitivity experiment, products chosen may include ERA-I or 5 (European
Reanalysis-Interim, or Version 5; Dee et al., 2011), NCEP-NCAR (National Center for
Environmental Prediction –National Center for Atmospheric Research; Kalnay et al., 1996),
JRA55 (Japanese 55-year Renanalysis; Kobayashi et al., 2015), and CFSR (Climate Forecast
System Reanalysis, Saha et al., 2014). Resolution will be coarsened to the lowest resolution
and temporal frequency will be chosen by the lowest temporal frequency available amongst
all the various products for the necessary variables, (listed in Table 3).

**4.2.2 Climate change catalogues**



For climate model resolution studies, CAM5 (Community Atmosphere Model, Version 5; Neale et al., 2010) 20th century simulations available at 25, 100, and 200 km resolutions from the C20C+ (Climate of the 20th Century Plus Project) Sub-project on Detection and Attribution (portal.nersc.gov/c20c) is available for participants to create AR catalogues for a

5 period of 27 years (1979-2005).  For climate change studies, high resolution (25 km) historical (1979-2005) and end of the century RCP8.5 (2080-2099) CAM5 simulation data are also provided.  This version of CAM5 uses the finite volume dynamical core on a latitude-longitude mesh (Wehner et al., 2014) with data freely available at portal.nersc.gov/C20C.

We use high resolution data for both the Tier 1 (~50km) and Tier 2 (25km) climate change catalogues because it has been shown that high resolution data is important in replicating AR climatology and regional precipitation. Although some climate models have a tendency to overestimate extreme precipitation related to ARs, these biases tend to decrease when high

15 resolution is applied (Hagos et al., 2015, Hagos et al., 2016). In an Earth system modelling framework, regional precipitation is represented more realistically in the higher resolution version compared to the standard lower resolution horizontal grids, (Delworth et al., 2012, Small et al., 2014, Shields et al., 2016).  High resolution data will have a better representation of topographical features and be better able to represent regional features at a finer scale.

### 4.3.3. CMIP5 catalogues

A number of studies have analyzed CMIP5 model outputs to explore future changes in ARs and the thermodynamic and dynamical mechanisms for the changes (e.g. Lavers et al., 2013;

25 Payne and Magnusdottir, 2015; Warner et al., 2015; Gao et al., 2016; Shields and Kiehl, 2016b, Ramos et al., 2016b). However, there is a lack of systematic comparison of the results



and how differences in AR detection and tracking may have influenced the conclusions regarding the changes in AR frequency, AR mean and extreme precipitation, spatial and seasonal distribution of landfalling ARs, and other AR characteristics, impacts, and mechanisms. Characterizing uncertainty in projected AR changes associated with detection

algorithms will facilitate more in-depth analysis to understand other aspects of uncertainty related to model differences, internal variability, and scenario differences, and such uncertainties influence our understanding of AR changes in a warming climate.

**5 Metrics**

Once all the catalogues are complete, then analysis will begin. There are many metrics to potentially analyze that are currently found in the literature. The frequency, duration, intensity, climatology of ARs and their relationship to precipitation are common. Other metrics, such as those described in Guan and Waliser (2017) can be adapted for ARTMIP. To

test the experimental design, we conducted a 1-month "proof of concept" test to help the basic design and fine tune a few metrics. Here we present a few results from this one month test that diagnose frequency, intensity and duration for two landfalling AR regions, the North Pacific and North Atlantic. For the full catalogues in Tier 1, additional regions will be analyzed, including the East Antarctic, which has proven to have large differences between

methodologies that implement a global algorithm compared to a regionally specific polar algorithm (Gorodetskaya et al., 2014). February 2017 was chosen because of the frequent landfalling North Pacific ARs during this time. Algorithms participating in the 1-month test are labelled with a $^+$ in Table 1 and identified with an algorithm id, i.e, A1, A2), etc. We also conducted a "human" control, where AR conditions and tracks were identified by eye for the

month of February for landfalling ARs impacting the western coastlines of North America and Europe. Full details on the human control dataset are explained in supplemental material.



We emphasize here that the human control is not considered "truth", but merely another
(subjective) method to add to the spectrum of detection algorithms participating in ARTMIP.

### 5.1 Frequency

Fig. 3 shows frequency (in 3-hour instances) by latitude band for landfalling ARs. The human
control as well as each of the methods are plotted for February 2017. Each color represents a
unique detection algorithm, and the black lines represent the human controls where both IVT
and IWV were utilized to identify ARs "by eye".  The IVT threshold (solid black line) is 250
kg m$^{-1}$s$^{-1}$ and the IWV thresholds (two different dashed lines) are 2 cm and 1.5 cm,
respectively. For western North America, all of the algorithms and the human controls agree
on the shape of the latitudinal distribution with most AR 3-hour-period detections
accumulating along the coast of California. ARs over the North Atlantic are latitudinally
more diverse, but the majority of algorithms and controls peak around 53$^{\circ}$N. Regarding the
actual number of 3-hour periods, there is a large spread in the frequency values across all the
automated algorithms with the human control "detections" far exceeding most algorithms.
This preliminary result suggests that setting a moisture threshold of 250 kgm$^{-1}$s$^{1}$ or an IWV
value of 2cm for North Atlantic ARs, as in the human control, is potentially too permissive.

To help identify case study events, a methodology count of how many (and which) methods
detect an AR along the coast can be conducted. Fig. 4 plots the number of methods that detect
an AR at the North American coastline for a sample of days in February, 2017. The number
of methods detections for each 3-hour time instance per day was computed, but only the
maximum time instance per day is plotted for simplicity.  The polygons represent the number



of methods. For example, if only 1 method detects an AR at a specific grid point along the coast, then a beige circle is plotted at that grid point along the coast; if 13 methods detect an AR at a specific grid point along the coast, then a dark blue circle is plotted at that grid point along the coast, and so forth. Even with this basic representation, the diversity in numbers of

method detections for each day is large. There are days where there is good method agreement in identifying AR conditions along the coastline. For example, February 7th, most methods identify AR conditions in Southern California, and on the 9th and 15th many methods detect ARs in the Pacific Northwest. However, there are many days where only a handful of methods detect ARs (i.e. February 22nd and 28th). The ability of individual algorithms to

detect the duration of events listed here is examined in further detail in Section 5.3.

**5.2 Intensity**

Intensity can be defined in many ways but often refers to the amount of moisture present in

an AR and/or the strength of the winds. IVT is an obvious quantity to use when evaluating the strength of an AR because it incorporates both wind and moisture.  There is value, however, at looking at these quantities separately when trying to decompose dynamic and thermodynamic influences. For the 1-month test, we looked at IVT for time instances where ARs exist.

In Figs. 5 and 6 we show two different ways of looking at mean AR-IVT across applicable methods to highlight how the definition of intensity can also vary. Fig. 5a/b show composites (for the North Pacific and European sectors, respectively) only at grid points where detection algorithms are implemented and include all time instances. This provides a look at the mean

IVT for all ARs at all locations for all times. Not all algorithms search for AR conditions at all points. For example, A14 (Shields and Kiehl) only detects ARs that make landfall along



coastal grid points, and A9 (Ramos et al.) detects ARs along reference meridians (for masks for regional algorithms, see Fig. S3 in supplementary material). Fig. 6 comparatively, shows IVT composites for each grid point, focusing only on specific time periods where landfalling ARs exist. While Fig. 5 shows mean IVT for all ARs at detection points, Fig. 6 is the

5 composite for landfalling ARs only. Each of these methods show intensity but are looking at different quantities. The landfalling ARs have a different signature and a less intense distribution, compared to the all-location AR composites. As one would expect, for both Figs. 5 and 6, methods with higher thresholds on IVT produce much higher AR average intensities, thus, AR intensity metrics could be thought of as self-selecting for some cases.

### 5.3 Duration

Duration of ARs also must be defined. Typically, this is expressed as the length of time an AR affects a point location, for example, a coastal location for a landfalling storm. However,

15 for tracking algorithms, duration may be defined as the life-cycle of an AR. For the 1-month proof-of-concept test, we use the first definition and look at the duration at coastal locations along the North American west coast and specific European locations. The top panel of Fig. 7 shows a time series of daily IVT anomalies along the western coastlines of the (orange line) Iberian Peninsula, (teal line) United States and (blue line) Ireland and the United Kingdom.

20 Four "human" observed AR tracks for events in each region are shaded and the composite magnitude of IVT for each are shown in panels a – d. These four events are compared over a variety of algorithms, indicated by algorithm id in the top panel, where each black dot indicates detection of an AR along the coastline. While all algorithms are listed, it is important to note that they are a mix of regional and global in scope.





The four selected events demonstrate the large diversity of AR geometry, landfall location and intensity that must be identified by each algorithm. The agreement between the different algorithms, hinted at in Fig. 4, is apparent in a comparison of the two West Coast examples mentioned in section 5.1 (Figs. 7b and d). The three versions of the Sellars et al. (2015)

algorithm can be used as a benchmark of AR intensity, in which the IVT threshold increases from 300 kg m$^{-1}$ s$^{-1}$ (in A11) to 700 kg m$^{-1}$ s$^{-1}$ (in A13). Relatively strong events are well captured by most algorithms (Fig. 7a - c), with few exceptions that are likely related to domain size. Agreement between algorithms on the duration or presence of an AR during weaker events is much more variable, such as that seen in Fig. 7d.

**5.4 Comparison with precipitation observational datasets**

The importance of understanding and tracking ARs ultimately boils down to impacts.  AR-related precipitation can be the cause of major flooding, can fill local reservoirs, and can

relieve droughts. How much precipitation falls, the rate at which it falls, and when and where it falls, specifically during AR events, is a metric we must consider for this project. The variation among the different algorithms can be seen in a comparison of precipitation characteristics for the event shown in Fig. 7b using MERRA-2 precipitation data  (Fig. 8) . The inset shows the landfalling mask from Shields and Kiehl (2016), which is used

as a common base of comparison for landfall between the different algorithms. Precipitation related to the landfalling AR is isolated by focusing only on gridboxes that are tagged by each algorithm. Comparison shows a positive relationship between the average spatial coverage of the detected landfalling plume (y-axis) and the average maximum precipitation rate at each time slice (x-axis). Generally, the durations of AR conditions along the coastline are higher

for algorithms with broader coverage.  The wide range of characteristics for this single well-defined event motivate further investigation.



As a part of Tier 1, methods will be evaluated using a variety of precipitation products in addition to MERRA-2, most relevant to the areas of interest. These include the Tropical Rainfall Measuring Mission (TRMM) Multisatellite Precipitation Analysis (TMPA) 3B42

product, version 7 (Huffman et al., 2007), the Global Precipitation Climatology Project (GPCP) dataset (Huffman et al., 2001), the Precipitation Estimation from Remotely Sensed Information Using Artificial Neural Networks (PERSIANN; Sorooshian et al., 2000), Livneh (Livneh et al., 2013), and E-OBS (Haylock et al., 2008). Tier 2 climate studies will use precipitation output, both convective and large-scale, from the CAM5 simulations. Finally, it

is important to consider not only the uncertainties in attributing precipitation due to detection method, but also the manner or technique used when assigning precipitation values to individual ARs.

## 6 Summary

ARTMIP is a community effort designed to diagnose the uncertainties surrounding atmospheric river science based on detection methodology alone. Understanding the uncertainties and, importantly, the implications of those uncertainties, is the primary motivation for ARTMIP, whose goals are to provide the community with a deeper understanding of AR tracking, mechanisms, and impacts for both the weather forecasting and

climate community. There are many detection algorithms currently in the literature that are often fundamentally different. Some algorithms detect ARs based on a condition at a certain point in time and space, while others follow, or track, ARs as a whole object through space and time. Some algorithms use absolute thresholds to determine moisture intensity, while others use relative measures, such as statistical or anomaly-based approaches. The many

degrees of freedom, in both detection parameter and choice of thresholds or geometry, add to

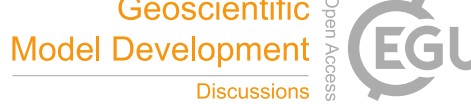

the uncertainty on defining an AR, in particular for gridded datasets such as reanalysis
products, or model output. This project aims to disentangle some of these problems by
providing a framework to compare detection schemes. The project is divided into two tiers.
The first tier is mandatory for all participants and will provide a baseline by applying all

algorithms to a common dataset, the MERRA-2 reanalysis. The second tier is optional and
will focus on sensitivity studies such as comparison amongst a variety of reanalysis products,
and a comparison using climate model data, utilizing both historical and future climate
simulations. Metrics diagnosed by ARTMIP will, at minimum, include AR frequency,
intensity, duration, climatology, and relationship to precipitation. Participation is open to any

group with an AR detection algorithm or an interest in evaluating ARTMIP data. Participants
will have full access to all ARTMIP data.

## 7 Data Availability

Data for ARTMIP is described in section 4. Source data applied to the 1-month proof of

concept test presented in this paper is available at the University of California, San Diego
(UCSD) Center for Western Weather and Water Extremes (CW3E) from B. Kawzenuk. Full
ARTMIP catalogues will be available to ARTMIP participants after the tier phases have been
completed. Participation in ARTMIP is open to any person or group with an AR detection
scheme and/or interest in analyzing data produced by ARTMIP. To do so, contact C. Shields

(shields@ucar.edu) or J. Rutz (jonathan.rutz@noaa.gov).

## 8 Acknowledgments

The contributions from NCAR (Cooperative Agreement DE-FC02-97ER62402), PNNL, and

LBNL to ARTMIP are supported by the U.S. Department of Energy Office of Science
Biological and Environmental Research (BER) as part of the Regional and Global Climate





Modeling program. PNNL is operated for DOE by Battelle Memorial Institute under contract DE-AC05-76RL01830. LBNL is operated for DOE by the University of California under contract number DE-AC02-05CH11231. Computing resources (ark:/85065/d7wd3xhc) were partially provided by the Climate Simulation Laboratory at NCAR's Computational and

5  Information Systems Laboratory, sponsored by the National Science Foundation and other agencies, as well as Scripps Institute for Oceanography at the University of California, San Diego. We also thank CW3E (Center for Western Weather and Extremes) for providing support for the 1$^{st}$ ARTMIP Workshop, and the many people and their sponsoring institutions involved with the ARTMIP project.

**9 Competing Interests**

Paul Ullrich is a topical editor of GMD, otherwise, the authors declare that they have no conflicts of interest.




**Figure 1:** Schematic diagram illustrating the diversity on AR detection algorithms found in current literature by categorizing the variety of parameters used for identification and tracking, and then listing different types of choices available per category.

| Parameter Type | Computation Type | Geometry Requirements | Threshold Requirements | Temporal Requirements | Regions (Examples) |
|---|---|---|---|---|---|
| | **Condition**<br>If conditions are met, then AR exists for each time instance at each grid point.<br><br>This counts time slices at a specific grid point. | Length | **Absolute**<br><br>Value is explicitly defined. | **Time slice**<br><br>Consecutive time slices can be counted to compute AR duration, but it is not required to identify an AR. | Global |
| Parameters Choices | | Width | | | North Pacific Landfalling |
| | | | **Relative**<br><br>Value is computed based on anomaly or statistic. | | North Atlantic Landfalling |
| | | Shape | | **Time stitching**<br><br>Coherent AR object is followed through time as a part of the algorithm. | Southeast U.S. |
| | **Tracking**<br>Lagrangian approach: if conditions are met, AR object is defined and followed across time and space. | Axis or Orientation | **No thresholds**<br>(object only) | | South America |
| | | | | | Polar |



**Figure 2**: Examples of different algorithm results. (Left and center) The fraction of total cool-season precipitation attributable to ARs from Dettinger et al. (2011) and Rutz et al. (2014). (right) As in (left and center), but for annual precipitation from Guan and Waliser (2015). These studies use different AR identification methods, as well as different

5   atmospheric reanalyses and observed precipitation data sets.

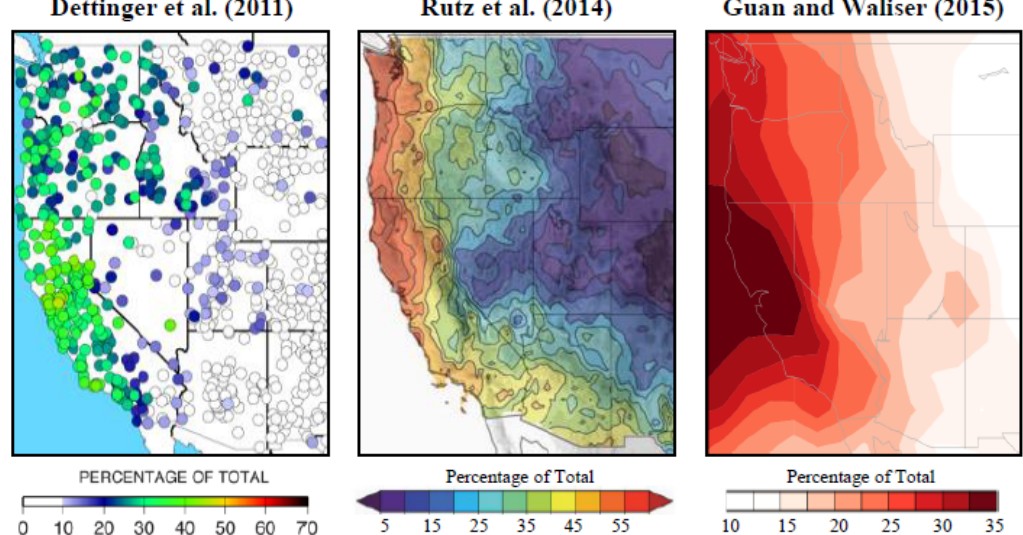



**Figure 3**: Human control vs method counts (3-hour instances) at the coastline for landfalling ARS by latitude for the month of February using MERRA-2 3-hourly data. West refers to North Pacific ARs making landfall along Western North America, and East refers to North Atlantic ARs impacting European latitudes. Color lines represent detection algorithms and

5 black lines represent the "human" control. The black solid line represents a static IVT 250 $kgm^{-1}s^{-1}$ threshold, and the black dashed (and dotted) lines represent static 2 and 1.5 cm IWV thresholds, respectively. Algorithm identifiers (A1, A2, etc.) specified in Table 1.

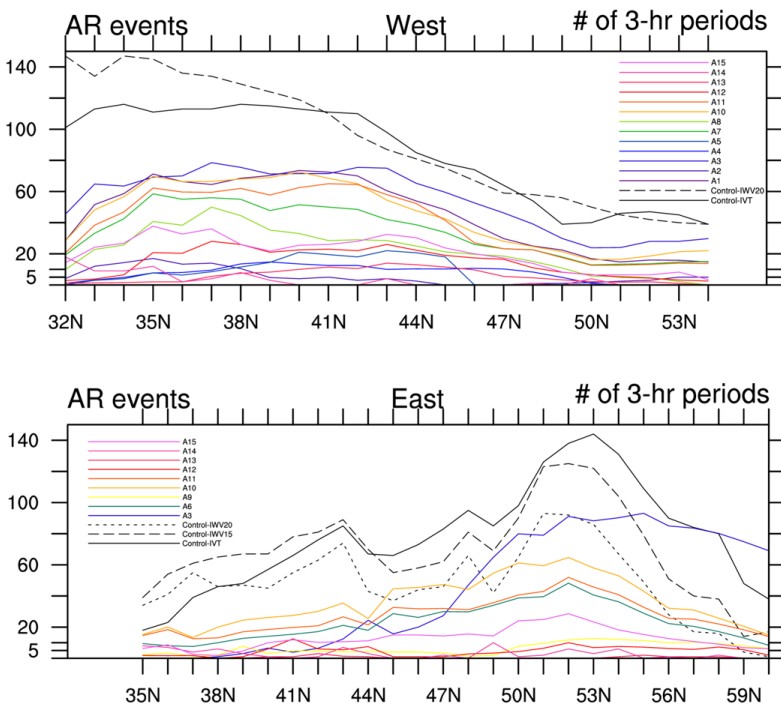



**Figure 4**: The number of methods that detect an AR at the coastline for sample days in
February is plotted, plots are labeled with date in YYYYMMDD, i.e. 20170201 is February
1st, 2017. Because each day had 8 associated time steps, the maximum number of methods
for each day is plotted. The polygons represent the number of methods, i.e. if only 1 method
5    detected an AR at a specific grid point along the coast, then a light beige circle is plotted at
that gridpoint along the coast; if 12 methods detected an AR at a specific grid point along the
coast, then the darkest blue star is plotted at that grid point along the coast. Individual
methods are not identified.

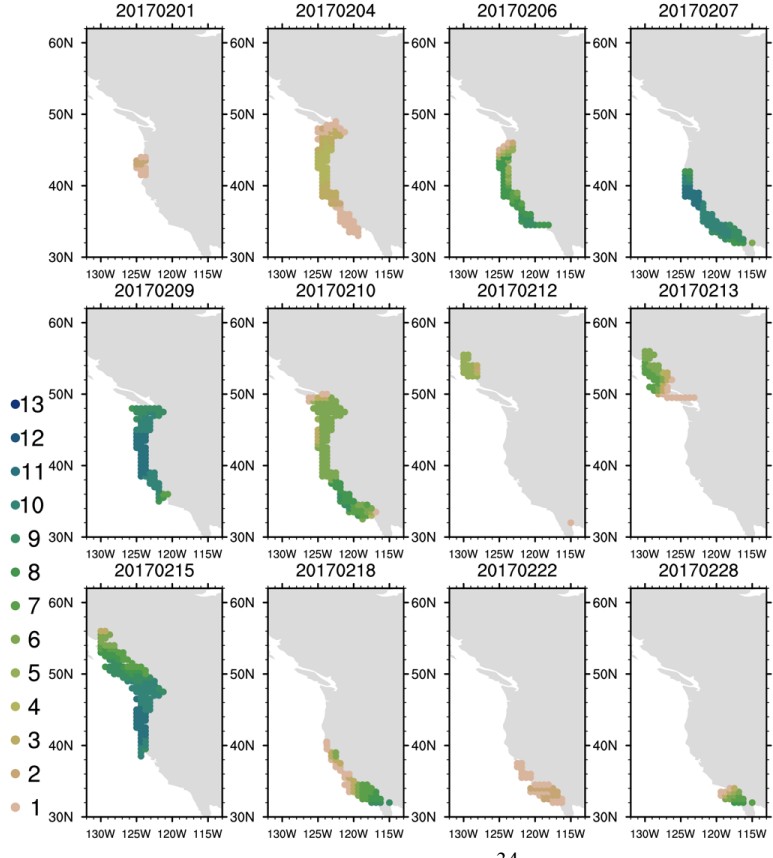



**Figure 5a** Composite MERRA-2 IVT (kg m$^{-1}$s$^{-1}$) for Western North America for all AR occurrences for all grid points where ARs are detected.  Algorithm ids are found in Table 1. Algorithm A14 computes AR detection only for landfalling ARs at coastline grid points. The absence of color indicates no AR detection.

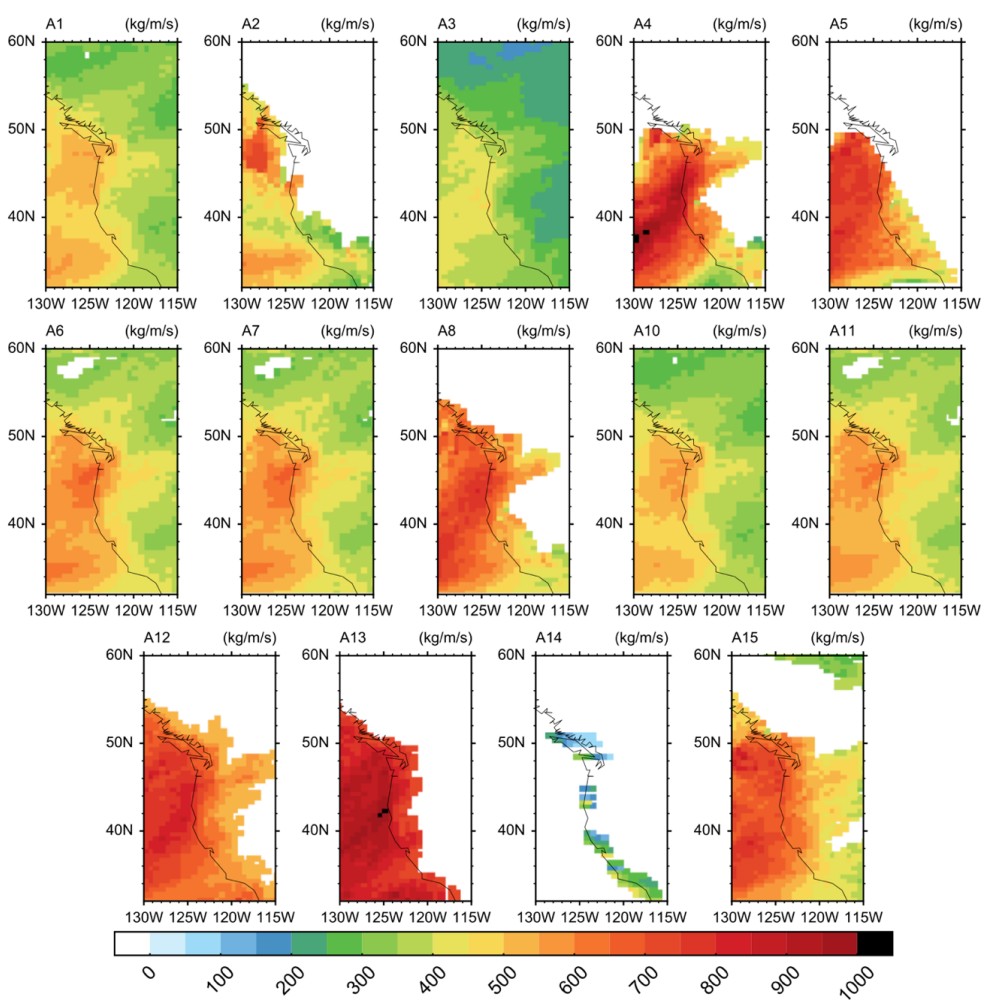



**Figure 5b**: Same as a except of North Atlantic ARs. Algorithm A9 detects ARs at reference meridians.

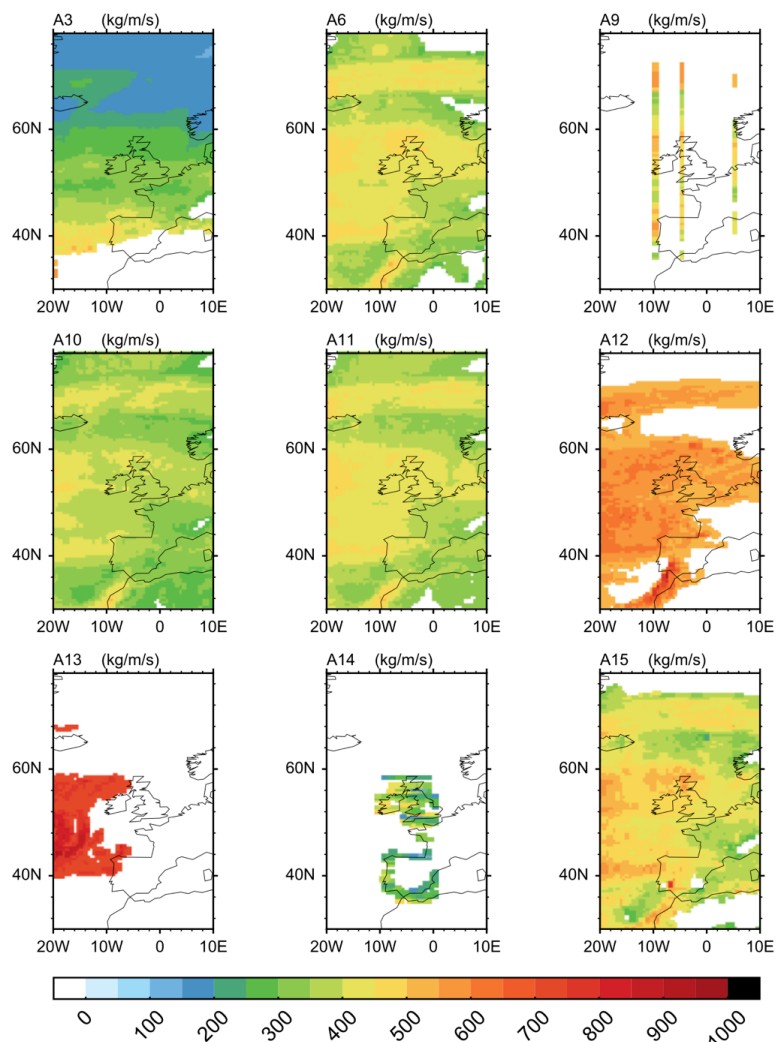





**Figure 6a** Composite MERRA-2 IVT (kgm$^{-1}$s$^{-1}$) but for landfalling ARs only along North American west coast. Time instances where an AR was detected along the coastline where composited for the entire region. Algorithm masks are not necessary.

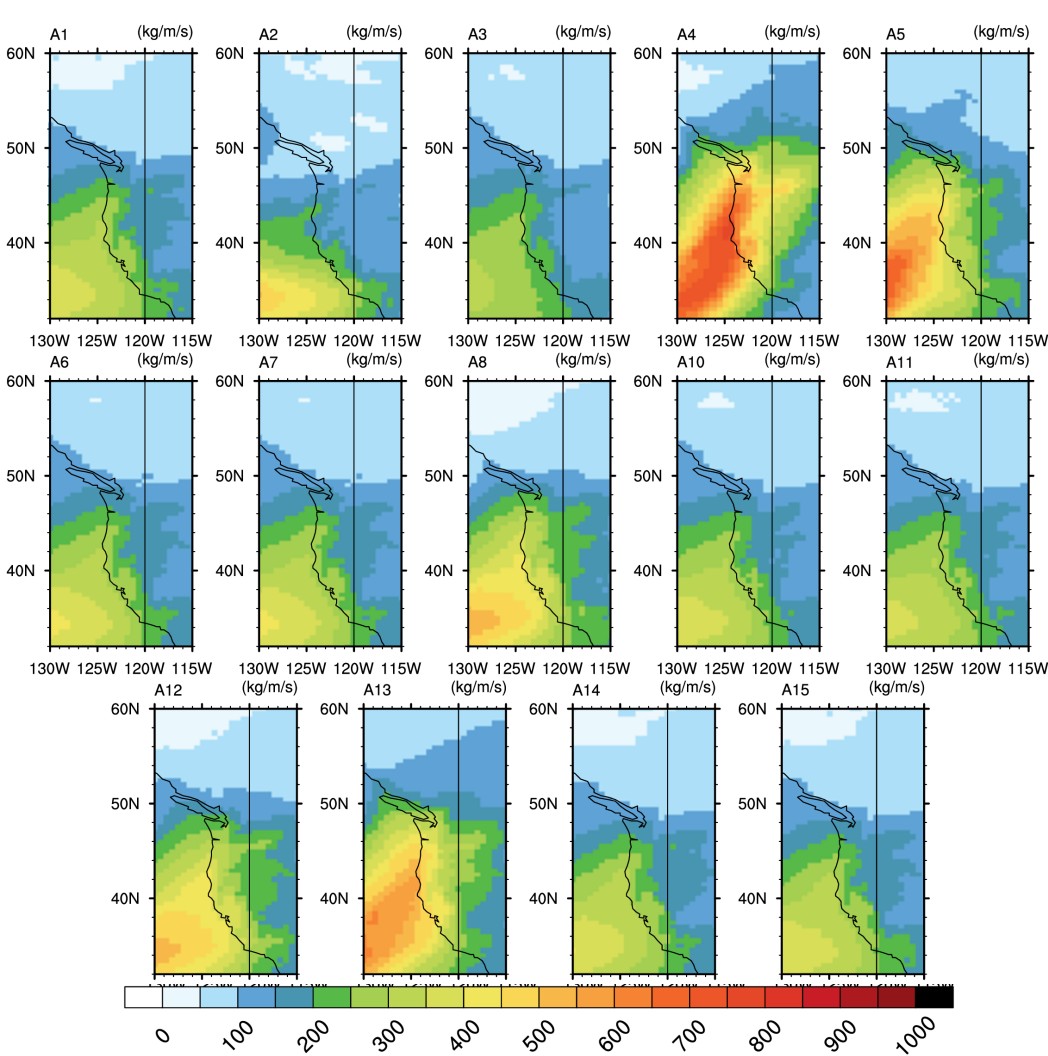

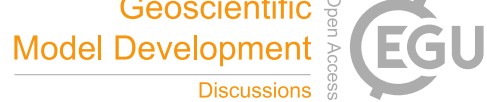



**Figure 6b:** As in a but for European coastlines.

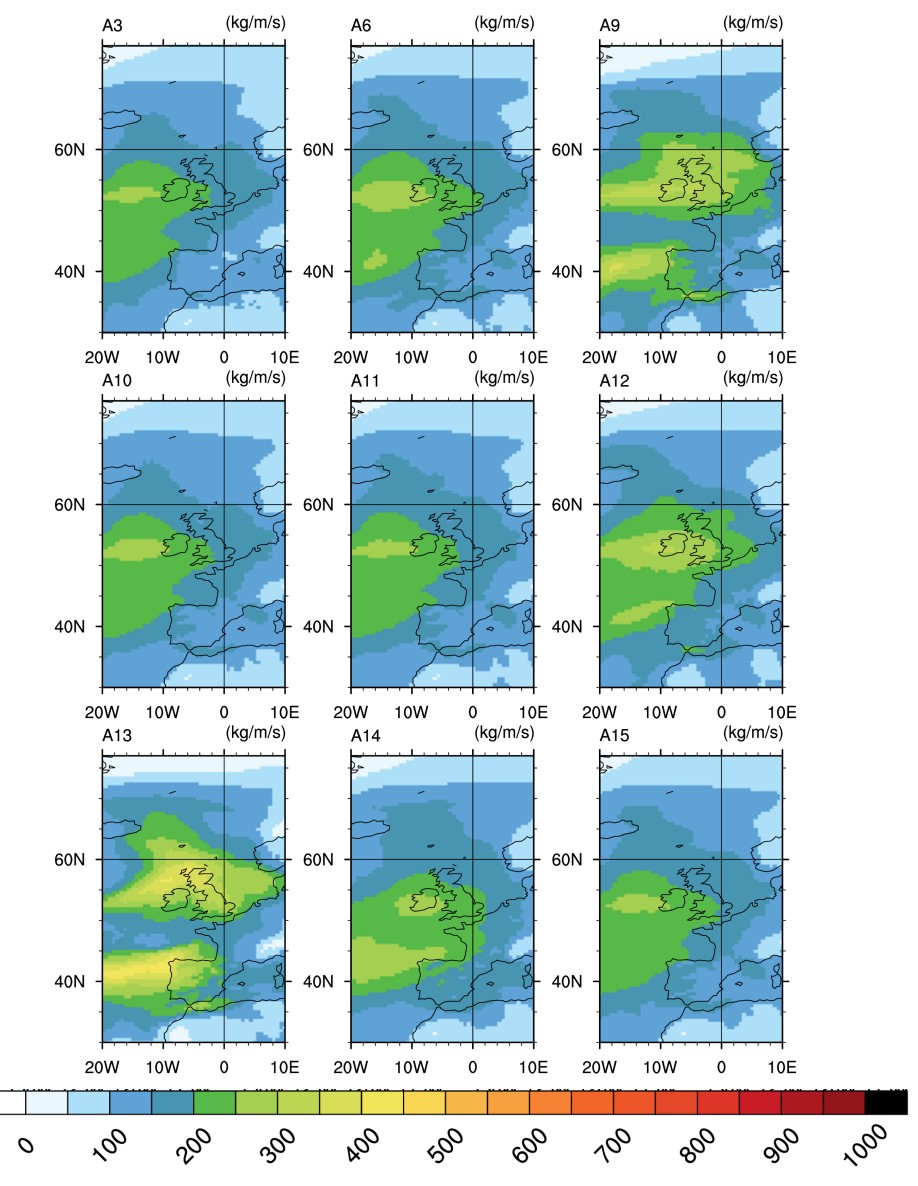





**Figure 7**: (top panel) Time series of daily IVT anomalies for (orange) Iberia, (teal) the U.S. West Coast and (blue) Ireland and the UK. Four events of varying geometry and intensity are shaded in the top panel and composites for each event are shown in the bottom four panels a -

5    d. The black dots above the time series in the top panel indicate time slices in which each event is detected by an algorithm.

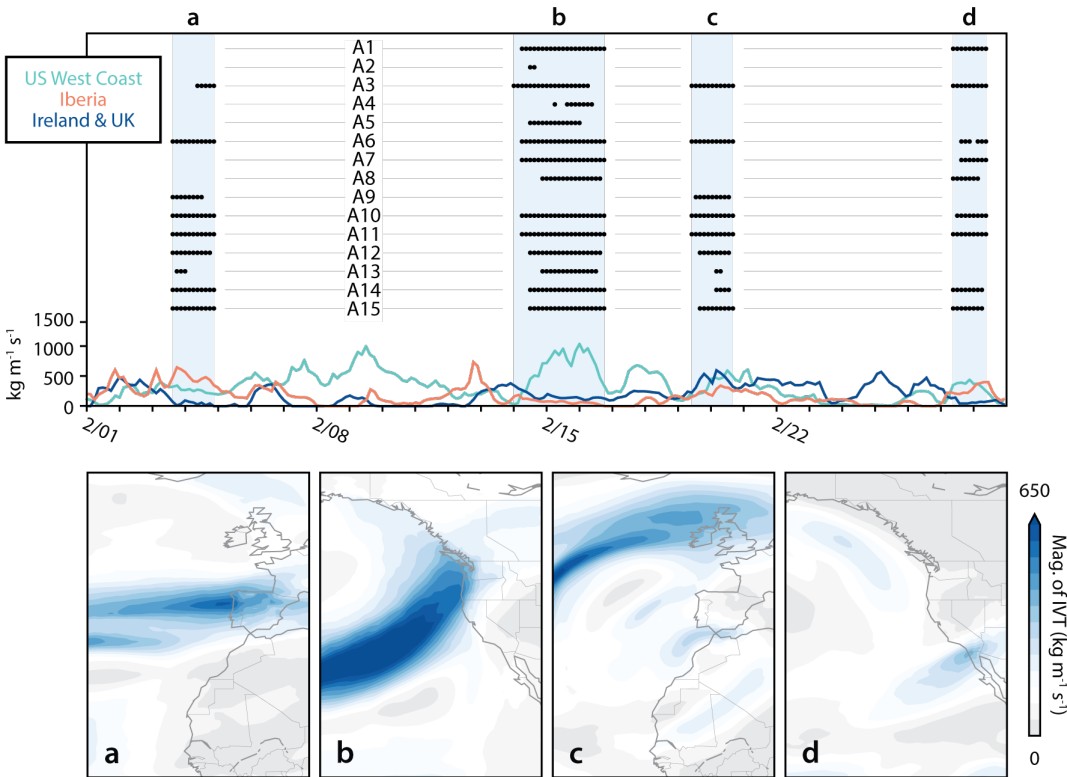



**Figure 8**: Focusing on the landfalling event in Fig. 7b, the average areal extent of the landfalling plume (y-axis) and average of the maximum precipitation rate at each detected time slice (x-axis) are compared for each algorithm. The size of the markers corresponds to the duration of the event as described in Fig. 7.

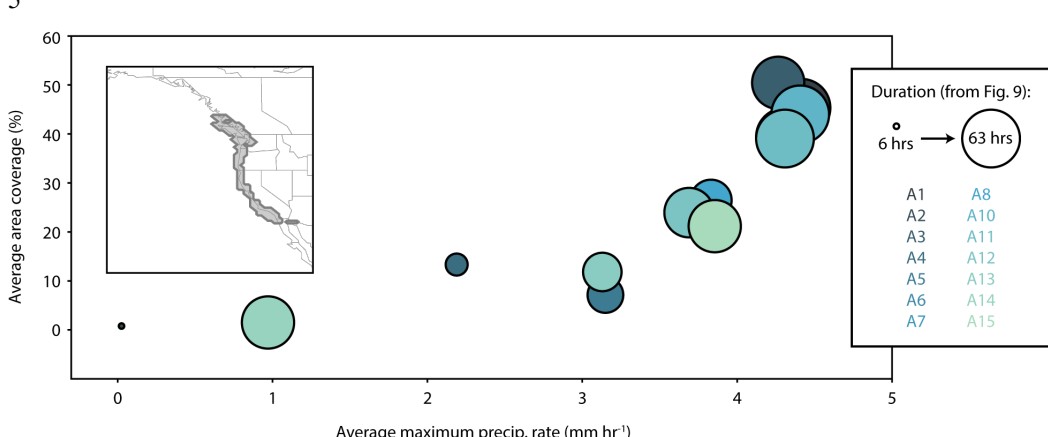



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
