# Peer review of "Atmospheric River Tracking Method Intercomparison Project (ARTMIP): Project Goals and Experimental Design"

_Geoscientific Model Development, 2017_

## Referee Comment (RC1) · Anonymous Referee #1 · 5 Mar 2018

Review of Shields et al.: Atmospheric river tracking method intercomparison project (ARTMIP): Project goals and experimental design

The paper of Shields et al. introduces the atmospheric river (AR) intercomparison project to the the community. The project aims at comparing different methods of identifying so called atmospheric rivers which are essential for moisture transport from the tropics to high latitudes. It uses different methods which have been used used so far in literature to identify atmospheric rivers, which are associated to cyclones in the extratropics.

The central goals of the project are to provide a neutral framework for comparisons of

ARs and to identify and quantify differences resulting from different AR methods, these are important to estimate humidity transport changes in a future changes. The basic quantities which are used are integrated water vapor transport (IWT) and integrate water vapor (IWV). Reanalysis data from ERA Interim, ERA-5 JRA-55 , NCEP-NCAR and CFSR are planned to be included in the comparison. For climate model studies CAM5 C20C+ in resolutions from 25 - 200 km in the historical period are used as well as the CMIP5 catalogue.

For a proof of concept one months test case will be analyzed (February 2017) and it is planned to extend the methods to larger sets of simulations.

I think the attempt to compare and synthesize different methods and metrics for the identification of ARs is indeed a very valuable and important effort. I wondered if and how systematic model related properties are treated for the evaluation, e.g which role plays horizontal and vertical , temporal resolution for the results and the applicability of the metrics? Is it possible to add a global (or at least northern hemispheric) view for the test period? Atmospheric rivers also do appear away from the continents and it would be interesting to see, how the algorithms identify these events on a hemispheric view.

I therefore recommend the paper after the following points have been addressed.

General points: Though the comparison of methods is essential and required to judge results of larger model simulations I missed the link to observations. How do the algorithms do compare with observations of atmospheric water vapor columns e.g. from satellite observations? Wouldn't it be useful to define criteria (or refine the definition of) ARs on the basis of satellite observations to allow to estimate the capability of algorithms and methods to identify the structures? Is it planned to give recommendations of methods to be used to identify ARs? How will region specific methods (i.e. only applicable to the Western U.S.) implemented for global analyses?

Technical: Figures: Why are captions placed above the Figures? This is confusing.

Fig.5a) and Fig.5b) (also Fig.6): Please label the color bar with units in both cases. Caption Fig.5b): Clarify text: "Same as Fig.5a) ..."

Please mention in the caption that the number of cases is different in Fig.5a) and Fig.5b) (also 6) due to the regional constraint of the respective definitions.

p.23, l.23: Why do not all algorithms participate in the 1-month test case?

Table 1: instead of using symbols (+,ˆ) for footnotes I suggest to use capital letters, which facilitates reading. Similarly explanations of *ZN and AR_coeff: Which methods refer to these quantities and what is the meaning of AR_coeff? Why is it 0.3 and is this a general number?

---

## Referee Comment (RC2) · Anonymous Referee #2 · 17 Apr 2018

article graphicx url

[Figure]

**Review of the Manuscript "Atmospheric River Tracking Method Intercomparison Project (ARTMIP): Project Goals and Experimental Design" by Christine A. Shields et al.**

Anonymous Referee

April 17, 2018

**1   General Comments**

This article comprises an overview of the experimental design and some initial results of the "Atmospheric River Tracking Method Intercomparison Project (ARTMIP)". Since the number of methods meant to detect and track atmospheric rivers has risen exponentially during the last few years, whereas, from my point of view, the term "atmospheric river" is still defined in a very rough manner, such an effort is urgently needed. The paper is well written and to the point, the experimental design and the aims of the project are described clearly. Therefore, I would like to recommend publication after addressing the minor comments listed below.
**2 Minor comments**

- Concerning the "Threshold Requirements" in Table 1, it would be very important to know whether the relative thresholds (normally percentiles) have been calculated separately for each month or season, for the winter half-year or for the entire year and I would suggest to include this information at this point. To provide an example, a method relying on IVT percentile thresholds calculated separately for each month or season will —by definition— produce a much higher number of ARs in the warm season than another method relying on all-year or winter half-year thresholds, given that the same percentile is considered. A higher number of ARs would on the one hand increase the coincidence rate with extreme precipitation events (which could be seen as some kind of "hit rate"), which is desirable, but on the other also increase the "false alarms", which is not desirable. Thus, a method could be "tuned" to fit some specific purpose and this could be discussed if you wish. Related to the above mentioned thoughts, the temporal window of the threshold calculation directly affects the seasonal cycle of the AR frequency counts, which is more pronounced for the "all-year" or "winter half-year" option and less so for the monthly/seasonal one. This holds for all variables associated with the seasonal cycle, i.e. also for the seasonal cycle of precipitation attributable to ARs.

- Section 3.2, page 17, lines: 9-13, "...but future climate research may be better served by relative methodologies, partly because of the model biases in the moisture and/or wind fields...": To circumvent the problem of using absolute thresholds in climate model output, you could calculate the percentile corresponding to a given absolute threshold (e.g. an IVT of 250 $kg\ m^{-1}\ s^{-1}$) in observations/reanalysis data and find the absolute value corresponding to this percentile in the historical run of the model. This absolute value would then also be used in the RCP run or this model.

[Figure]

- Section 4.2.1: You could also consider to use the NOAA-CIRES 20th Century Reanalysis and/or ECMWF's ERA-20C to have AR presence-absence time series for the entire 20th century, but this is of course just a suggestion. Doing so, you could e.g. assess aspects of low frequency variability associated with the PDO or AMO.

- Section 5.1, page 24, lines 19-20: Here you state that "a moisture threshold of [...], as in the human control, is potentially too permissive". However, from my point of view, the human control should be always better than any automated method so it is the methods having a problem at this point, not the human eye. Since several persons observing the same IVT field could come to distinct conclusions on whether an AR is present or not, the rough AR definition you cite on page 6 (the AMS one) should be still improved to come to a better consensus on what an AR actually is. Anyway, from my point of view, it is wrong to claim that the human eye is worse than the automated methods.

- Section 5.4 on page 27 and Figure 8: I would recommend to use an independent "purely observational" precipitation dataset other than MERRA.

- Figure 4: I would suggest to use a discrete instead of continuous colorbar for this figure.

- Figures 5 to 6: Adding "IVT" below or next-to the colorbar would be helpful in these figures.

- Caption of Figure 6a: A space is missing after "kg" in the parenthesis.

---

## Author Comment (AC2) · 18 Apr 2018

Response to GMDD Interactive comment from Referee #2

We would like to thank the reviewer for these constructive comments. We will address each point below:

Q: Concerning the "Threshold Requirements" in Table 1, it would be very important to know whether the relative thresholds (normally percentiles) have been calculated separately for each month or season, for the winter half-year or for the entire year and I would suggest to include this information at this point.
A: This is a great suggestion. Although not all "relative" methods are the same in terms of relying on climatology, for the methods that do, we will amend the Table, or add supplementary material, to include this information.

Q: Section 3.2, page 17, lines: 9-13, "...but future climate research may be better served by relative methodologies, partly because of the model biases in the moisture and/or wind fields...": To circumvent the problem of using absolute thresholds in climate model output, you could calculate the percentile corresponding to a given absolute threshold (e.g. an IVT of 250 kg môĂĂĂ1 sôĂĂĂ1) in observations/ reanalysis data and find the absolute value corresponding to this percentile in the historical run of the model. This absolute value would then also be used in the RCP run or this model.

A: Yes, this method of identifying which threshold to use for future runs would work well. Algorithmic choices are left up to the developers themselves, depending on the science question of interest. Comparing and contrasting the different algorithm choices is a priority for ARTMIP.

Q: Section 4.2.1: You could also consider to use the NOAA-CIRES 20th Century Re-analysis and/or ECMWF's ERA-20C to have AR presence-absence time series for the entire 20th century, but this is of course just a suggestion. Doing so, you could e.g. assess aspects of low frequency variability associated with the PDO or AMO.

A: We will consider and discuss these datasets our upcoming workshop, where we will discuss the reanalysis Tier 2 catalogues.

Q: Section 5.1, page 24, lines 19-20: Here you state that "a moisture threshold of [...], as in the human control, is potentially too permissive". However, from my point of view, the human control should be always better than any automated method so it is the methods having a problem at this point, not the human eye. Since several persons observing the same IVT field could come to distinct conclusions on whether an AR is present or not, the rough AR definition you cite on page 6 (the AMS one) should be still improved to come to a better consensus on what an AR actually is. Anyway, from

my point of view, it is wrong to claim that the human eye is worse than the automated methods.

A: This is certainly a subject for debate within the community as well as ARTMIP. We do not want to imply that automated methods are better than human controls, or vice versa, and did not intend to give this message. We will amend the language to be more neutral.

Q: Section 5.4 on page 27 and Figure 8: I would recommend to use an independent "purely observational" precipitation dataset other than MERRA.

A: This is a good recommendation. We intend to use several different datasets for Tier 1 and Tier 2 analysis. For this paper, and the proof-of-concept analysis, we used the MERRA-2 as a first look, only given this is only one month of data. The intent for this analysis is to show that our design functions properly and to display the types of metrics we will delve into more deeply for full ARTMIP catalogues.

Q: Figure 4: I would suggest to use a discrete instead of continuous colorbar for this figure.

A: We will adjust the colorbar.

Q: Figures 5 to 6: Adding "IVT" below or next-to the colorbar would be helpful in these figures.

A: Thank you for catching this omission. We will correct it.

Q: Caption of Figure 6a: A space is missing after "kg" in the parenthesis.

A: Thank you for catching this syntax problem. We will correct it.
* * *

---

## Author Response (AR1)

*Referee Comments are in red text with italics*. Author responses are in black text. New additions to the manuscript, where applicable, are noted *in italics*.

**Response to GMDD Interactive comment from Referee #1**

Questions to answer in overview:

*I think the attempt to compare and synthesize different methods and metrics for the identification of ARs is indeed a very valuable and important effort. I wondered if and how systematic model related properties are treated for the evaluation, e.g which role plays horizontal and vertical, temporal resolution for the results and the applicability of the metrics? Is it possible to add a global (or at least northern hemispheric) view for the test period?*

Thank you for your review and noting the importance of ARTMIP.  During the course of Tier 1, ARTMIP participants will be disentangling these great questions.

We will absolutely include a global view in a Tier 1 overview paper, which we are actively working on now. However, for this GMDD paper (and the results from the 1-month test we conducted last Fall), the purpose is twofold: 1) document the experimental design for the community and outline the goals for the project, and 2) act as a proof-of-concept that such an intercomparison will work and produce interesting results.  Given that we are actively working with the full algorithm catalogues (i.e, the 1980-2017 MERRA-2 period), we feel that we can better use our resources and serve the community by presenting the full results in our science overview paper, rather than add an additional plot with a global view to this experimental design paper.  The figures we present here are intended to show the flavor of the types of analysis we are doing but are not necessarily meant to be final "results". We expect some of these metrics may change after looking at the full Tier 1 data, especially so given that have a larger pool of catalogues for Tier 1 is available compared to those analyzed for the proof-of-concept.  We have added a comment to the paper to make clear that we intend to present global metrics in future research.  We have also included a global snapshot of MERRA-2 IVT data with sample ARs labelled in the supplementary material to illustrate example events across the globe.

In terms of temporal resolution, we begin to address the issue in the supplemental text (Figures S1 and S2) where we show the difference between MERRA-2 3 hourly derived data versus 1-hourly data.  As for horizontal resolution, all algorithms are applied to the same MERRA-2 data (~50km) for Tier 1, and 25km data for Tier 2-climate change.  Note that other Tier 2 plans include intercomparing different reanalysis products at different resolutions.

*Additions to manuscript*: New text can be found on pages 23 and 27 in the updated manuscript. The new figure is in the supplemental material, Figure S4. We chose February 19th 21Z to tie into Figure 7 on duration where each algorithm detection tag is shown by the black dots.

General points:

*Though the comparison of methods is essential and required to judge results of larger model simulations I missed the link to observations. How do the algorithms do compare with observations of atmospheric water vapor columns e.g. from satellite observations? Wouldn't it be useful to define criteria (or refine the definition of) ARs on the basis of satellite observations to allow to estimate the capability of algorithms and methods to identify the structures?*

We have actually spent a great deal of time thinking about how we might perform a verification of sorts, but it turns out to be much easier said than done. The AMS Glossary of Meteorology recently adopted a definition for atmospheric rivers, a definition vetted by numerous subject matter experts and commented on during town hall meetings at major conferences. This definition is very broad, and while some generic numbers are provided via schematics, no exact criteria for identifying or tracking an AR are described. The upshot is that while there is broad consensus among the community regarding very general characteristics of ARs, there is no agreement regarding the variables, magnitudes, or precise geometries that should be used to identify them. So, while we could, for example, assess which methods most closely track satellite observed IWV features > 2000 km long and < 500 km wide (one definition, among many), we would already be self-selecting our preferred algorithm. This is true across a host of definitions. In light of this, we have chosen to focus our project on quantifying the uncertainty that arises as a result of these various algorithms and making this data available, at which point other researchers can feel free to pursue additional analyses.

*Is it planned to give recommendations of methods to be used to identify ARs?*

We will need to wait for the results, but we have some ideas about the form some of these recommendations might take. For example, certain algorithms might produce outliers in terms of AR climatology over certain regions. We may have reason to trust some algorithms more than others over certain regions, or to trust some algorithms more than others in anticipating the effects of a changing climate on ARs. We will need time to work out these details. What we do not want to do is rank algorithms. Recommendations will be based on identifying algorithm(s) that are most appropriate for specific science questions.

*How will region specific methods (i.e. only applicable to the Western U.S.) implemented for global analyses?*

All ARTMIP participants are running their algorithms for the North American west coast, the European west coast, and globally, or in whichever of those regions their algorithm is capable of running. We also have several algorithms being applied to Polar regions. We expect that algorithms developed specifically for a given region will be in closer agreement with each other than those developed for other regions and then applied there, but we don't yet have those results. To compare *all* algorithms (i.e. global with regional) we need to look at the common

denominator. For example, we have looked at landfalling ARs on the US West Coast because all catalogues include this area. However, this does not preclude sub-setting algorithms for other areas.  Note that European landfalling ARs is a subset of the total number of algorithms in ARTMIP.  Metrics applicable to global-only algorithms can also be analyzed in isolation.  We also note that Tier 1 catalogues will be made available to the community after the completion of this phase and hope that members of the community will take advantage of the catalogues.

*Additions to manuscript:* We have added text to the manuscript to outline why and how we compare all algorithms (page 23 in the updated manuscript).

Technical Comments:

We have addressed all the technical comments. The point by point comments are below.
*Figures 5a/b, 6a/b, and Table 1 have been modified as well as the addition of the Newman et al. reference to Table 1 footnotes.*

*Why are captions placed above the Figures?*
Agreed, we have moved captions to below the figure.

*Fig.5a) and Fig.5b) (also Fig.6): Please label the color bar with units in both cases. Caption Fig.5b): Clarify text: "Same as Fig.5a) ..."*
We have incorporated these suggestions into the revised manuscript.

*Please mention in the caption that the number of cases is different in Fig.5a) and Fig.5b) (also 6) due to the regional constraint of the respective definitions.*
We have noted this explicitly in the figure captions.

*p.23, l.23: Why do not all algorithms participate in the 1-month test case?*
The reason all algorithms are not represented in the 1-month test is simply that we had a schedule, and not all participants' data was available at that time (we have also added new participants for Tier 1 and Tier 2 since the posting of this GMDD paper).

*Table 1: instead of using symbols (+,ˆ) for footnotes I suggest to use capital letters, which facilitates reading. Similarly, explanations of *ZN and AR_coeff: Which methods refer to these quantities and what is the meaning of AR_coeff? Why is it 0.3 and is this a general number?*

We have changed the symbols to letters. To avoid confusion with A1, A2, etc. and the notation for the algorithms, we will use lower case lettering "a", "b", and "c."

*AR_coeff* refers to an empirically-derived coefficient used to define atmospheric rivers in the foundational paper of Zhu and Newell (1998).  More work and a deeper exploration of this

value was also done by Newman et al (2012). Methods that use (or are based on) the Zhu and Newell method of AR identification are noted by footnotes and are: the Gorodetskaya et al. and Shields and Kiehl methods. The Zhu and Newell paper is already cited in the references, but we have added the Newman paper to the footnote in Table 1 for those interested in more detail.

Zhu, Y., and R. E. Newell (1998), A proposed algorithm for moisture fluxes from atmospheric rivers, Mon. Weather Rev., 126(3), 725–735, doi:10.1175/1520-0493(1998)126<0725:APAFMF>2.0.CO;2.

Newman, M., G. N. Kiladis, K. M. Weickmann, F. M. Ralph, and P. D. Sardeshmukh (2012), Relative contributions of synoptic and low-frequency eddies to time-mean atmospheric moisture transport, including the role of atmospheric rivers, J. Clim., 25, 7341–7361.

**Response to GMDD Interactive comment from Referee #2**

We would like to thank the reviewer for these constructive comments. We will address each point below:

*Concerning the "Threshold Requirements" in Table 1, it would be very important to know whether the relative thresholds (normally percentiles) have been calculated separately for each month or season, for the winter half-year or for the entire year and I would suggest to include this information at this point.*

This is a great suggestion. Although, it is important to note that not all "relative" methods are the same in terms of relying on climatology. For the methods that do, footnotes [c] in Table 1 explain what each percentile-method used for climatology.

*Additions to manuscript*: We have expanded the footnotes section to be clearer regarding how climatologies are computed by the algorithms that use percentiles to define thresholds.

*Section 3.2, page 17, lines: 9-13, "...but future climate research may be better served by relative methodologies, partly ecause of the model biases in the moisture and/or wind fields...": To circumvent the problem of using absolute thresholds in climate model output, you could calculate the percentile corresponding to a given absolute threshold (e.g. an IVT of 250 kg m 1 s 1) in observations/ reanalysis data and find the absolute value corresponding to this percentile in the historical run of the model. This absolute value would then also be used in the RCP run or this model.*

Yes, this method of identifying which threshold to use for future runs would work well. Algorithmic choices are left up to the developers themselves, depending on the science

question of interest. Comparing and contrasting the different algorithm choices is a priority for ARTMIP.

*Section 4.2.1: You could also consider to use the NOAA-CIRES 20th Century Reanalysis and/or ECMWF's ERA-20C to have AR presence-absence time series for the entire 20th century, but this is of course just a suggestion. Doing so, you could e.g. assess aspects of low frequency variability associated with the PDO or AMO.*

We have discussed these datasets our recent workshop, where we will discuss the reanalysis Tier 2 catalogues. The 20[th] Century Reanalysis is being considered but the timeline for this decision has been pushed back. The high-resolution climate change and the CMIP5 catalogues will precede the reanalysis catalogues in Tier 2.

*Additions to manuscript*: We have added the 20[th] Century NOAA-CIRES for consideration, page 23 in the updated manuscript.

*Section 5.1, page 24, lines 19-20: Here you state that "a moisture threshold of [...], as in the human control, is potentially too permissive". However, from my point of view, the human control should be always better than any automated method so it is the methods having a problem at this point, not the human eye. Since several persons observing the same IVT field could come to distinct conclusions on whether an AR is present or not, the rough AR definition you cite on page 6 (the AMS one) should be still improved to come to a better consensus on what an AR actually is. Anyway, from my point of view, it is wrong to claim that the human eye is worse than the automated methods.*

This is certainly a subject for debate within the community as well as ARTMIP. We do not want to imply that automated methods are better than human controls, or vice versa, and did not intend to give this message. We have amended the language to be more inclusive.

*Additions to the manuscript:* We have added additional language (in the metrics section) to be more neutral concerning human controls vs automated methods. We chose to add it here, rather than the results section, because we feel this statement should be made when human controls are first discussed. See page 24 in the updated manuscript.

*Section 5.4 on page 27 and Figure 8: I would recommend to use an independent "purely observational" precipitation dataset other than MERRA.*

This is a good recommendation. We intend to use several different datasets for Tier 1 and Tier 2 analysis. For this paper, and the proof-of-concept analysis, we used the MERRA-2 as a first look, only given this is only one month of data. The intent for this analysis is to show that our design

functions properly and to display the types of metrics we will delve into more deeply for full ARTMIP catalogues.

*Figure 4: I would suggest to use a discrete instead of continuous colorbar for this figure.*

We have adjusted the color bar to apply more discrete colors as suggested.

*Figures 5 to 6: Adding "IVT" below or next-to the colorbar would be helpful in these figures.*

Thank you for catching this omission. We have corrected this, as per the request of another referee.

*Caption of Figure 6a: A space is missing after "kg" in the parenthesis.*

Thank you for catching this syntax problem. This has been corrected.

[revised manuscript text omitted]

**Moved (insertion) [1]**

[Figure]

Figure S4. Integrated vapor transport (IVT) (kg/m/s) for a sample time slice in the MERRA-2 dataset. Date stamp in the upper right corner in the format of YYYYDDMMHH, i.e. 2017, February 19th, 21Z. Note the various individual ARs in the Pacific and Atlantic Ocean Basins with landfalling ARs impacting the west coast of the United States, the UK, and mid-latitude coastal regions on the west coast of South America.